# Homologous Recombination and Repair Functions Required for Mutagenicity during Yeast Meiosis

**DOI:** 10.3390/genes14112017

**Published:** 2023-10-28

**Authors:** Liat Morciano, Renana M. Elgrabli, Drora Zenvirth, Ayelet Arbel-Eden

**Affiliations:** 1Department of Genetics, Hebrew University of Jerusalem, Jerusalem 91904, Israel; liat.morciano@mail.huji.ac.il (L.M.); renana.elgrabli@mail.huji.ac.il (R.M.E.);; 2The Medical Laboratory Sciences Department, Hadassah Academic College, Jerusalem 91010, Israel

**Keywords:** DNA double-strand breaks (DSBs), meiosis, mutations, DNA repair, homologous recombination

## Abstract

Several meiotic events reshape the genome prior to its transfer (via gametes) to the next generation. The occurrence of new meiotic mutations is tightly linked to homologous recombination (HR) and firmly depends on Spo11-induced DNA breaks. To gain insight into the molecular mechanisms governing mutagenicity during meiosis, we examined the timing of mutation and recombination events in cells deficient in various DNA HR-repair genes, which represent distinct functions along the meiotic recombination process. Despite sequence similarities and overlapping activities of the two DNA translocases, Rad54 and Tid1, we observed essential differences in their roles in meiotic mutation occurrence: in the absence of Rad54, meiotic mutagenicity was elevated 8-fold compared to the wild type (WT), while in the *tid1Δ* mutant, there were few meiotic mutations, nine percent compared to the WT. We propose that the presence of Rad54 channels recombinational repair to a less mutagenic pathway, whereas repair assisted by Tid1 is more mutagenic. A 3.5-fold increase in mutation level was observed in *dmc1∆* cells, suggesting that single-stranded DNA (ssDNA) may be a potential source for mutagenicity during meiosis. Taken together, we suggest that the introduction of de novo mutations also contributes to the diversification role of meiotic recombination. These rare meiotic mutations revise genomic sequences and may contribute to long-term evolutionary changes.

## 1. Introduction

A central role of meiosis is to generate haploid gametes with new combinations of genetic material. Genetic variation is achieved by recombination and independent assortment of chromosomes. These meiotic events contribute to genome heterogeneity by scrambling and shuffling preexisting sequences. Rare, de novo mutations that occur during meiosis further increase variation and are the only source of new genetic information and novel alleles. About sixty years ago, Magni and Von Borstel (1962) [1] found that the rate of de novo mutation in *Saccharomyces cerevisiae* meiosis is higher than the rate in mitosis, and they coined the term “the meiotic effect” [2]. Since their discovery, we and others have confirmed that the rates of meiotic mutation are elevated in yeast cells [3,4,5] and in humans [6,7]. Depending on the locus examined, the mutation rates in meiosis are 6- to 22-fold higher than in mitotically growing cells [1,5].

Several lines of evidence link meiotic mutations to the recombination process: An early finding was that colonies with de novo mutations at *his4* showed a high frequency of recombination of markers bracketing the gene [8]. Likewise, meiotic mutations at the *CAN1* reporter had a high frequency of recombination between two adjacent markers, covering 2.2 kb around this reporter, compared to non-recombinant colonies or compared to vegetatively growing cells [5]. Meiotic mutations were found to depend strongly on DNA double-strand breaks (DSBs), induced by the meiosis-specific endonuclease Spo11. We further showed that most meiotic mutations do not occur during S-phase, which is the phase where most DNA synthesis takes place [5]. Rather, elevated meiotic mutagenicity appears at the same time and by the same kinetics as recombination events [5]. These findings imply that breakage, followed by recombination, is an intrinsic factor enhancing de novo mutagenicity. Nonetheless, we found that approximately half of the meiotic mutations were not associated with recombination events between closely flanking markers [5], and we therefore suggested that mutations may result not only from recombinational repair but also from sister chromatid repair of meiotic DSBs [9]. This leaves an open question: What are the relative contributions to meiotic mutagenicity of the various repair functions that operate during meiosis? We address this question in the current study.

The meiotic recombination process in the yeast *S. cerevisiae* is well characterized, with an orderly sequence of events in which many genes participate. Meiotic recombination is initiated by massive, tightly controlled breakage of the DNA via the Spo11 endonuclease and its assisting factors [10,11,12,13]. Next, the MRX (Mre11/Rad50/Xrs2) protein complex plays an essential role as a DNA damage sensor, and the resulting DNA ends are digested to create single-strand 3′ DNA ends (ssDNA), which are ~820 nts long [14]. Following end-resection, strand invasion is mediated by two recombinases, Rad51p and the meiosis-specific Dmc1p [15]; both act concurrently to generate interhomolog bias during the HR process [16,17]. The presence of the Rad51 protein, but not its strand invasion activity, is required for proper Dmc1-mediated strand invasion [18]. Rad51’s action requires interactions with Rad54, a DNA translocase, which assists in strand invasion [19,20] and stimulates Rad51 to preferentially utilize the sister chromatid for repair [21,22,23]. Likewise, the Rad54 paralog Tid1/Rdh54 is needed to stimulate Dmc1 activity to bind recombinogenic sites [24], and together, Tid1 and DMC1 preferentially act to promote interhomolog recombination [23]. In normal meiosis, the Hed1 protein inhibits Rad51-mediated recombination [25,26] by competing with Rad54 [27]. The direct phosphorylation of both Rad54 and Hed1 by Mek1 [28] prevents Rad51/Rad54 complex formation [29,30]. The invaded intermediates are substrates for DNA synthesis and the subsequent ligation of joint molecules (JMs). These JMs generate several types of recombination intermediates that mainly resolve to either non-crossover (NCO) or crossover (CO) products.

In the present study, we focus on the HR process, which is a core event enhancing meiotic mutagenicity, to better define recombinational repair functions required for mutagenicity during meiosis. We generated a series of mutants deleted in genes involved in meiotic recombination (namely *SPO11*, *MRE11*, *DMC1*, *HED1*, *RAD54*, *TID1*, and *MUS81*) and examined their effects on the occurrence of meiotic mutations. Combining mutant analysis and time course experiments, we revisited the meiotic HR repair pathway to study the kinetics of meiotic mutagenicity, recombination, and the interactions between these processes. Our results indicate that some HR genes are involved in mutagenic DSB repair pathways and others are involved in repair pathways that reduce mutagenesis. ssDNA that is exposed prior to JM formation and the various ways by which DSB repair is handled contribute to generate balanced meiotic mutagenicity that, on one hand, provide genome variability and, on the other hand, do not overload the genome with excessive deleterious variants.

## 2. Materials and Methods

### 2.1. Strain Construction

All strains used for this study are of the SK1 genetic background and are listed in Table 1. Gene deletions, insertions, and gene replacements were obtained using standard polymerase chain reaction (PCR)-based methods.

The original “founder” strains for all strains generated in this study are OMY78-1 and OMY75-2 [5]. A detailed description and a thorough characterization of the wild-type (WT) strain is found in Mansour et al. (2020). In brief, we inserted a copy of the *ADE2* gene ~400 bps upstream to the functional copy of this hemizygous reporter and we selected for mutations in the functional *CAN1* allele, which lead to resistance to canavanine, by spreading ascospores on plates containing canavanine and lacking adenine (Can −Ade). To enable detection of intragenic recombination events, these strains also contained heteroallelic mutations in *HIS4* (*his4-X*/*his4-B*) [31], and recombination was measured by recovering colonies on −His plates.

Strains deleted for *MRE11*, *DMC1*, *HED1*, *HED1DMC1*, *TID1*, and *MUS81* were obtained by generating the relevant PCR-amplified gene product from the *kanMX4*-deletion library (http://chemogenomics.pharmacy.ubc.ca/hiplab/GGCN_Lab/SGDP/group/yeast_deletion_project/deletions3.html, accessed on 20 September 2023) and the transformation of this PCR fragment into the haploid *YRA10* strain, followed by further crosses and ascus dissections.

*SPO11* was deleted in the haploid strain by transformation with the corresponding PCR-amplified product from the hisG-*URA3*-hisG plasmid. *URA3* was “popped out” by growing the transformed colonies on 5FOA.

The *RAD54*-deleted strain (ALY92) was obtained by haploid *rad54∆::URA3* strain with YRA12 and selecting the desired Ura+ spore colony following tetrad dissection.

The *RAD54*-deleted strain was manipulated to carry the *cdc7-as3* allele (ALY170). Haploids of the latter were obtained by mating the *RAD54 cdc7-as3* haploid strain OMY78-1 with ALY92 (*rad54Δ*), followed by tetrad dissection. The *cdc7-as3* allele was detected by PCR.

### 2.2. Media

YPD (complete medium with glucose as a carbon source), YPA (complete medium with acetate as a carbon source, used for pre-sporulation), SPM (sporulation medium), and SPO (solid sporulation medium) were prepared as described previously [32]. The drop-out media (various amino acid and nucleotide mixes) were prepared as described by Rose et al. (1990), except that lysine, leucine, tryptophan, arginine, and adenine were added at concentrations of 80 mg/L each, when required. Can −Ade medium: canavanine (Sigma C9758) was added at 60 mg/L to the synthetic complete medium lacking arginine and adenine. SCLG-2 medium: synthetic complete [33], with 0.5% glucose and without adenine and arginine.

### 2.3. Determining Mutation Rates during Mitotic Cell Divisions with Fluctuation Analysis

Yeast strains were thawed from frozen stocks and streaked to single colonies on YPD plates. In each fluctuation assay, single colonies from the appropriate strains were patched on YPD plates, and after overnight growth at 30 °C, the patches were lightly replica-plated onto the selective medium, Can −Ade (60 mg/L). After 4 days, areas with no mutant colonies were chosen, and cells from the corresponding YPD plate were cultured in liquid SCLG-2 medium [34]. Cultures were shaken overnight at 30 °C, diluted into SCLG-2 medium, and dispensed into 96-well plates, 100 µL per well. When testing the *CAN1* mutation rate, ~1000 cells were cultured per well. Cultures in 96-cell plates were grown for 48 h at 30 °C without shaking in a high humidity incubator, to prevent evaporation. For each culture, six wells were pooled and diluted, and cells were counted using a Neubauer counting chamber. To determine the Nt value, which reflects the actual number of viable cells per subculture (well) at the end of the experiment, 100 µL aliquots of the pooled and diluted cultures were plated on each of 4 YPD plates. Then, 100 µL of double-distilled water was added to each of the remaining 90 subcultures. For each subculture, 100 µL was spot-plated onto dried Can −Ade plates to identify mutant colonies (9 spots per plate). Plates were allowed to dry for 3 h at room temperature and then incubated at 30 °C for 5 days, after which the number of spots without resistant colonies was recorded.

The method chosen to calculate the mutation rate in these mitotically growing cultures was the P-zero (P_0_) method [35]. As the number of mutations per culture follows the Poisson distribution, the proportion of cultures without mutant colonies, P_0_, is the zero term of the Poisson, given by the following equation [36]:$$a=-\frac{ln(p0)}{Nt}$$
where *a* is the mutation rate, *p*0 is the fraction of subcultures (spots) with zero colonies, and *Nt* is the average number of cells per subculture at the time of application of the selective agent.

### 2.4. Determination of Meiotic Mutation and Recombination Rates

To determine mutation and recombination rates during and after meiosis, cells from −80 °C stocks were streaked to single colonies onto YPD plates. Several single colonies were then picked and patched (~2 cm × 2 cm patch) onto another YPD plate and incubated overnight. This plate was replica-plated onto three different plates: SPO, −His (to exclude preexisting recombinants), and Can −Ade (to exclude preexisting mutants). The patch that showed the highest sporulation efficiency on the SPO plate was chosen, and a clean section that did not show any colonies on the remaining two plates (−His and Can −Ade) was marked. A culture of each relevant strain, from the corresponding YPD plate, was inoculated into 2 mL liquid YPD, grown overnight at 30 °C, and then resuspended in 100 mL of liquid YPA at a dilution of 1:2500, followed by incubation with vigorous shaking (200 rpm) at 30 °C for ~18 h to reach a titer of 1–2 × 10^7^ cells/mL. In cases where strains grew more slowly (such as in *mre11Δ* and *rad54Δ* strains), the cells were resuspended in YPA at a dilution of 1:1000. Cells were then washed twice in water and resuspended in the same volume (100 mL) of liquid SPM, which was pre-warmed to 30 °C. Upon resuspension, a sample was taken and, after appropriate dilutions, spread on the selection plates (−His or Can −Ade); this represented the meiotic time zero sample. The remaining culture was then vigorously shaken (200 rpm) at 30 °C. The volume of the shaking flasks was ten times greater than the volume of the liquid medium (YPA or SPM) to allow efficient aeration. Cells were taken for analysis after 2–8 h in sporulation conditions and spread on selection plates, which were then incubated at 30 °C for 2–7 days, after which recombinant and mutant colonies were counted. The number of colonies appearing on selection plates from time zero was always subtracted from the numbers obtained at later time points during meiosis, as the time zero colonies reflect events that occurred in the mitotic divisions prior to meiosis. To determine the number of cells that had been spread on the selection plates, serial 1:10 dilutions were made, until we obtained a cell suspension of ~5 × 10^3^ cells/mL (as determined by counting the cells in a Neubauer counting chamber). Three aliquots of 100 µL each were then spread on YPD plates to score for cell viability at each time point. To deduce the original number of cells in each sample, the average number of colonies that grew on YPD plates was multiplied by the dilution factor. Recombination and mutation rates were then calculated. Sporulation efficiency after 24 h in sporulation conditions was determined microscopically by counting monads, dyads, and 3–4-spore asci vs. non-sporulated cells.

### 2.5. Meiotic Time Course Experiments

Meiotic time course experiments were performed to follow the kinetics and rates of recombination and/or mutation and to monitor cell viability along the meiotic process.

In time course experiments, the same protocol was followed as described above. Usually, 4 samples were collected after 0, 4, 6, and 8 h in sporulation medium (SPM), and whenever needed, a 2 h sample was also collected. Cells were diluted in 100 mL of liquid YPA and resuspended in the same volume of pre-warmed SPM in 1 L flasks. Cell samples were collected and treated as explained above.

In earlier studies using similar meiotic conditions, we showed that in our SK1 strains, global DNA synthesis at S-phase takes place at around 2 h after transfer to SPM, DSBs begin to appear at 3 h, recombination occurs between 4 and 6 h, and meiosis is completed after about 8 h [37,38]. Diploid *dmc1Δ* cultures embark on meiosis but become arrested in meiotic prophase I [39] and do not reach the haploidization commitment point. Return to mitotic growth (RTG) rescues the viability of these *dmc1Δ* cultures and allows one to assess the levels of recombination and mutations, although the cultures remain diploid.

### 2.6. Meiotic Experiments Using the Inhibitor PP1

For strains bearing the *cdc7-as3* allele, meiosis was induced in the presence of the inhibitor PP1 [4-amino-1-tert-butyl-3-(p-methylyphenyl) pyrazolo [3,4-d] pyrimidine], which halts cells after meiotic S-phase [40]. Following thawing from −80 °C and patching as described above, cells were inoculated in 2 mL liquid YPD, grown overnight (at 30 °C), and then resuspended in 60 mL of liquid YPA at a dilution of 1:2500 for the *RAD54* strain, or of 1:1000 for *rad54Δ*, followed by incubation with vigorous shaking (200 rpm) at 30 °C for ~18 h to reach a titer of 1–2 × 10^7^ cells/mL. Cells were then washed twice in water and resuspended in 60 mL and divided into two flasks of liquid SPM. Upon transfer to SPM, PP1 was added to half the culture at a final concentration of 15 µM. The other half was left untreated to proceed normally through meiosis. After 8 h in sporulation conditions, both PP1-treated and untreated cultures were spread on YPD plates, as well as on selection plates (as described above), and rates of recombination and mutation were calculated.

### 2.7. Statistical Treatments

Bars of mutation rates in the figures represent standard errors (SE), calculated from n independent experiments with each strain, performed on different days. Most meiotic experiments were based on at least 8 independent experiments. In the WT, we based our analytical calculations on at least 19 meiotic experiments. Comparisons of mutation rates between strains were based on t-tests (at 0.05 level).

## 3. Results

De novo meiotic mutations lead to genetic variability by generating novel alleles. To date, much of our knowledge regarding de novo mutations derives from the analysis of somatic mutations appearing during mitotic cell divisions in normal and in cancerous cells [41]. In the present work, we focus on mutations that occur during meiosis, which show a marked enhancement compared to mitotic mutations, and study the occurrence of meiotic mutagenesis in relation to recombination functions.

We first characterized the parental reference wild-type (WT) diploid strain (YRA23-1), which shows rapid and synchronous meiosis upon transfer to SPM, as described in the Materials and Methods and [37,38]. YRA23-1 showed a high sporulation efficiency (~90%) and maintained 100% and 80% viability 8 h and 24 h following the induction of meiosis, respectively. Spores recovered after tetrad dissection were 96% viable (366 viable spores out of 380, Table 2).

In order to measure the frequency of mutagenesis, we used the reporter gene *CAN1*; mutations in this gene enable growth on canavanine medium (Can −Ade) [2,34]. Since these mutations are recessive, we constructed a diploid strain in which *CAN1* is expressed only from one copy of chromosome V, in the SK1 genetic background [4,5]. Thus, mutations in the single functional copy of *CAN1* allow growth on canavanine medium. To follow recombination around the *CAN1* reporter, we inserted the *ADE2* gene upstream of the functional *CAN1* allele and the gene *URA3* downstream of the promoterless *can1* allele (see Materials and Methods). Colonies that could grow without adenine and without uracil could only have been produced by crossing over within the 2210 bp spanning the *CAN1* reporter [5]. Our yeast strains are also heterozygous for two separate alleles of the *HIS4* gene (*his4-X*/*his4-B*; Table 1) to enable the detection of intragenic (heteroallelic) recombination events; only recombinant products could grow on minimal medium that lacked histidine [31]. The meiotic mutation and recombination rates for WT YRA23-1 are shown in Table 2 and were used as the baseline reference. In WT cells that were transferred to SPM (aliquot taken at time zero in meiosis), a frequency of *can1*-mutated colonies of ~6.65 × 10^−7^ was recorded. This number of colonies, appearing on Can −Ade plates at the onset (time zero) of all experiments, was subtracted from the numbers obtained at each later time point during meiosis, as the time zero colonies reflect events that occurred prior to meiosis. Thus, all graphs show zero mutation rate at the X axes at the beginning of the time course experiment and show only the net meiotic mutation rates.

### 3.1. Meiotic Mutations Depend on the Generation of DSBs

***spo11Δ* and *mre11Δ* strains:** As we and others have shown, de novo mutations during meiosis firmly depend on *SPO11*-induced DSBs [4,5]. A null mutation in *MRE11* also eliminates meiotic recombination (reviewed in reference [42]). To confirm the dependence of meiotic mutations on DSB formation, we analyzed meiotic mutagenicity in *spo11Δ* and *mre11Δ* strains. Strains homozygous for either *spo11∆* or *mre11Δ* were induced into meiosis (see Materials and Methods for details), and aliquots were taken at the time of initiation of meiosis and 8 h afterwards. The rates of mutagenesis and recombination were determined and compared to the WT strain, YRA23-1 (Table 2). In a diploid strain homozygous for *spo11Δ*, cell viability stayed high until 4 h into meiosis [5] and gradually decreased afterwards, reaching 36% at 8 h (Table 2). The *spo11Δ* strain showed no increase in the rate of recombination at *HIS4* during meiosis compared to WT. When meiotic mutagenicity was assessed in the *spo11* mutant, only a few mutated colonies were observed at 8 h (~10% of the mutations accumulated in the WT strain (Table 2 and [5]). Cell viability in diploids homozygous for *mre11* dropped to 56% 8 h after induction of meiosis (Table 2). Similar to *spo11Δ* cells, no recombination was observed at *HIS4* in the *mre11Δ* strain during meiosis (Table 2).The mutation rate observed at *CAN1* in *mre11*-deleted cells was ~20% of the WT level at 8 h (Table 2). Hence, the dependence of meiotic mutations on DSBs was confirmed not only in cultures bearing the *spo11* deletion but also in *mre11Δ* cells.

### 3.2. Recombinases and Proteins Involved in Strand Exchange Affect Meiotic Mutagenicity

Both the Rad51 recombinase and the meiosis-specific Dmc1 recombinase play roles in meiotic DSB repair. Rad51 catalyzes DNA strand exchange and is essential for both mitotic and meiotic HR [43]. Yeast mutants that are deleted for *RAD51* show low viability, poor recombination [44], and no DSB repair during meiosis [38]. Due to its poor meiotic phenotype, we did not examine *rad51Δ* cells directly; instead, we examined the involvement of gene functions that regulate the activity of Rad51 (and of Dmc1), namely those coded by *RAD54*, *TID1*, and *HED1*.

***dmc1Δ*, *hed1Δ***, **and *hed1Δ*
*dmc1Δ* strains:** We performed time course experiments in meiosis (see Materials and Methods) using strains homozygous for *dmc1Δ*, *hed1Δ*, or the double-mutant *hed1Δdmc1Δ*. Cell viability in *dmc1Δ* and *hed1Δ* mutants, and in the *hed1Δdmc1Δ* double mutant, remained high, around 100%, until 8 h following the induction of meiosis, similar to the WT strain (Figure 1A). The rates of mutagenesis in *CAN1* and recombination events at *HIS4* were examined in each strain and compared to the WT. Wherever possible, recombination was also recorded around the *CAN1* reporter itself (this is possible only when cells complete meiosis and form viable haploid products). In total, 84% of the cells had sporulated in *hed1∆* and in the *hed1Δdmc1Δ* cultures [25]. 93% of *hed1Δ* spores were viable, and 55% of the spores in the *hed1Δdmc1Δ* double mutant showed germination capacity, compared to 96% viable spores in WT cells (Table 2). Similar results were previously reported for other strains from the SK1 background; 96% in *hed1Δ* and 70% in the double mutant, compared to 96% in the WT [45].

Upon 8 h in meiosis, the rates of recombination at *HIS4* in *dmc1Δ* and in *hed1Δ* mutants were 43% and 49%, respectively, compared to the WT, while in the *hed1Δdmc1Δ* double mutant, recombination dropped to 12% (Figure 1B and Table 2). Our observation regarding recombination frequencies at the HIS4-LEU2 hotspot is consistent with previous findings of a 2-fold decrease in interhomolog bias in *hed1∆* diploids [16,45]. However, little effect on recombination levels was observed for another hotspot [46]. We could not measure recombination levels around *CAN1* in *dmc1Δ* cells since these cells arrest in meiosis and remain diploid following RTG [39]. In canavanine-resistant (*Can^R^*) colonies obtained from *hed1Δ* cells after 8 h in meiosis, we observed 40% recombination between the *CAN1*-flanking markers *URA3* and *ADE2* (69 Ura^+^ Ade^+^ colonies, out of 171 *Can^R^* haploid colonies examined). In *hed1Δ dmc1Δ* cells, the rate of recombination around *CAN1* was 25% (38 Ura^+^ Ade^+^ colonies, out of 153 *Can^R^* haploid colonies examined). These results are summarized in Table 3.

***rad54**∆* and *tid1**∆* strains**: Rad54 and Tid1 both operate as chromatin remodeling proteins during vegetative growth and in meiosis [22,47,48,49]. We therefore examined mutation rates during both mitotic and meiotic divisions. To determine mutation rates during vegetative growth, we conducted fluctuation experiments (as described in the Materials and Methods) in *rad54∆* and in *tid1∆* strains as well as in WT, using *CAN1* as a reporter gene (Table 4). In WT cells, we observed a rate of 0.2 mutations per million mitotic cell divisions, and in *tid1*-deleted cells, the mutation rate was 1.5-fold higher. The mitotic mutation rate in *rad54Δ* cells was ~26-fold higher than in the WT strain (Table 4).

Mutation and recombination rates during meiosis were determined by meiotic time course experiments in diploids homozygous for *rad54∆* or *tid1∆*. Cell viability gradually decreased during meiosis, reaching 84% viability for the *rad54∆* strain and 58% for the *tid1∆* strain at 8 h (Figure 2A).

Examination of recombination rates and kinetics at *HIS4* in *rad54Δ* cells showed levels and timing comparable to the WT (Figure 2B and [21]). In the *tid1Δ* strain, however, clear retardment and overall low recombinant levels of 35% compared to the WT were recorded (as previously observed [21]). Moreover, when mutation rates at the *CAN1* reporter were measured, a considerable difference between the *rad54Δ* and *tid1Δ* strains was observed: at 4 h, the mutation rates in *rad54Δ* cells and in *tid1Δ* cells were 7.1 and 0.1 mutations per 1 million cells (the WT with 0.27 mutations per 1 million cells). At later time points, the mutation rate in *rad54Δ* cells was higher than in the WT by 8-fold and reached 9.7 mutations per 1 million cells at 8 h (Figure 2C and Table 2). In *tid1Δ* cultures, however, mutation levels remained low, 0.09-fold of the WT level at 8 h (Figure 2C and Table 2). Interestingly, at the meiotic start point, when the time zero aliquot was taken, frequencies of 2.12 × 10^−5^ and 1.06 × 10^−8^
*Can^R^*-mutated colonies were recorded in *rad54Δ* and in *tid1Δ* cells, respectively. The *rad54Δ* strain showed a 32 times higher mutation level than the WT basal level (~6.65 × 10^−7^) at the same time point. As in all strains examined in this study, this value (which represents mutation events that had occurred prior to meiosis) was subtracted from the numbers obtained at each later time point during the *rad54Δ* meiotic time course.

### 3.3. Evaluation of Mutation Occurrence during Meiotic S-Phase in rad54∆ Cells

As described above, the *rad54Δ* strain showed a pronounced elevated meiotic mutation level, 8-fold higher compared to the WT (Table 2). As can be seen in the time course experiments that follow the kinetics of mutation enhancement in *rad54Δ* strains (Figure 2C), the increase in mutation levels is well observed at 4 h, but only minor (if any) elevation is detected at 2 h. In WT SK1 strains, such as YRA23-1, the S-phase takes place at around 2 h [37]. We tested the hypothesis that the meiotic S-phase is not the source for the elevated meiotic mutation levels observed in the *rad54Δ* mutant. To check this, we took advantage of the *cdc7-as3* mutation, which is a conditional allele of the protein kinase *CDC7* [40]. The Cdc7-as3 kinase is inactivated by the addition of the inhibitor PP1 [4-amino-1-tert-butyl-3-(p-methylyphenyl) pyrazolo [3,4-d] pyrimidine] to the sporulation medium; in the presence of PP1, cells homozygous for *cdc7-as3* complete DNA replication but arrest before DSB formation and remain diploids. The *cdc7-as3* mutant, therefore, is used as a tool for obtaining synchronous cell cultures [40] and for separating meiotic S-phase mutations from DSB-induced mutations during HR [5]. Strains bearing the *cdc7-as3* allele allow us to distinguish between mutations that occurred prior to recombination, in cells that underwent meiotic DNA replication (in the presence of PP1) and arrested (in the presence of PP1), and mutations in cells of the same strain that underwent both replication and recombination (without PP1). We previously confirmed that DNA replication is indeed completed in WT cells of the SK1 genetic background that are arrested upon exposure to PP1. This was achieved by FACS analysis to track the DNA content in WT cells undergoing meiosis and by visualizing the cells with microscopy, 8 h upon PP1 addition [5].

We introduced the *cdc7-as3* allele into the *rad54Δ* strain (ALY94) to obtain a diploid strain that is homozygous for both *cdc7-as3* and the *rad54Δ*-deleted gene (see Table 1 for full genotypes). This diploid (ALY170) and the corresponding WT strain, OMY93, were induced into meiosis in the presence or absence of PP1 (for details, see the Materials and Methods). Samples were spread on Can −Ade medium (as well as on YPD) upon transferring the culture to SPM to induce meiosis and after 8 h in SPM. Exposure to PP1 for 8 h did not affect cell viability in any of the strains (Figure 3A). As expected, after 8 h in SPM, meiotic cell cultures that contained PP1 showed no detectable recombination in WT *cdc7-as3* as well as in *rad54Δ cdc7-as3* cells, since cells that are treated with PP1 are arrested prior to prophase I with no DSBs [5,40]. Cells that were not treated with PP1 reached high levels of recombination in WT and in *rad54Δ cdc7-as3* cells (1.3 × 10^–2^ and 0.8 × 10^–2^, respectively, see also in Figure 2B). Meiotic mutation levels upon exposure to PP1 in WT cells were very low at the time of entry into meiosis and remained low after 8 h (5.2 × 10^–7^), which means that essentially no new mutated cells were generated during the meiotic S-phase in the WT culture, as we previously reported [5]. Similarly, in the *rad54Δ cdc7-as3* strain, very few *Can^R^* mutant colonies (2.3 × 10^–6^) were detected in cultures that were arrested with PP1 upon meiosis (Figure 3C). This indicates that the mutations that had accumulated during the meiotic process in WT and in *rad54Δ* cells did not result from replication failure or from mismatched bases introduced during the meiotic S-phase DNA synthesis. These experiments using the PP1 inhibitor complement the time course experiments described above (Figure 2). Taken together, the data suggest that the elevated mutagenicity observed in the *rad54Δ* strain results from the meiotic recombinational repair process that is triggered by DSBs.

### 3.4. The Involvement of Mus81 in Mutagenicity of Meiotic Joint Molecule Resolution

**The *mus81**∆* strain:** Mus81p endonuclease plays a role in joint molecule (JM) formation and resolution during meiosis in *S. cerevisiae* [50,51] and directs meiotic recombination towards interhomolog interactions [50]. To evaluate mutagenicity levels of JM resolution that depend on Mus81, we examined meiotic mutagenicity and recombination rates in *mus81∆* cells (Figure 2B,C). Following the induction of meiosis, the cell viability of *mus81∆* cells was maintained around 94% until 4 h and gradually dropped to 58% at 8 h. Recombination frequencies remained lower than in the WT throughout the experiment and reached 38% of the WT level after 8 h (Figure 2B and Table 2). Recombination levels were also measured in *Can^R^* colonies that were obtained 8 h into the meiosis of *mus81Δ* cultures. Of 229 *Can^R^* haploid colonies examined, 34 colonies showed the exchange of flanking markers (*URA3* and *ADE2)* around *can1*, which is 15% recombination (Table 3). Meiotic mutagenicity levels in *mus81Δ* cells were lower by ~30% of the WT level at 8 h (Figure 2C and Table 2) and reached 0.85 mutated colonies per 1 million cells.

## 4. Discussion

De novo mutations are found at elevated levels following meiotic recombination in yeast cells and are triggered by DSBs [4,5,8]. In humans, crossovers bear more de novo mutations than non-crossovers in male sperm [6] and mutations are increased 50-fold around crossovers, as observed with whole-genome sequencing of many human families [7]. These findings suggest that meiotic DNA repair by HR is inherently mutagenic. The molecular mechanism that contributes to the enhanced occurrence of new mutations in meiosis is not clear [52,53]. In the present study, we explore the effect of several recombinational repair genes on the occurrence of new mutations during meiosis. We examine not only the rates of mutagenicity at the meiotic endpoint but also the timing and kinetics of the appearance of new meiotic mutations in relation to recombination and, wherever relevant, compare meiotic mutagenesis to mutation rates obtained during mitotic cell divisions, as determined the same *CAN1* reporter gene.

DSB formation is a prerequisite for most meiotic mutations, as evident from the deleted strains of the *SPO11* or *MRE11* genes (Table 2). What other recombination functions affect mutagenicity?

Rad51 recombinase is essential for both mitotic and meiotic HR (reviewed by Bishop [54]), whereas Dmc1 is meiosis-specific [15]. Rad51 catalyzes recombination directly in mitosis and indirectly, via Dmc1, during meiosis [18], so that the interplay between these two recombinases leads to homologue bias repair during meiosis, as previously reviewed [17]. Since the deletion of *RAD51* causes cell death and severe impairment of recombination during meiosis [38,44] and the deletion of *DMC1* leads to meiotic arrest in *S. cerevisiae* [39], we chose to explore mutagenicity during the HR process by analyzing the involvement of genes that assist and regulate the prime recombinases without severe interruption of the recombination process, namely *RAD54*, *TID1*, and *HED1*. The corresponding recombination-defective deletion mutants proceed into meiosis, maintain reasonable viability, and generate (at least to some extent) recombinant products.

### 4.1. Rad54 Restrains Mutagenicity during Meiosis and in Mitotically Cycling Cells

High mutation rates are observed in mitotically cycling cells that are deleted of the *RAD54* gene (Table 4). Meiotic mutations also increased substantially in the *rad54∆* strain (~8-fold over the WT level), while the recombination rate and kinetics remained fairly similar to the WT (Figure 2B,C). As determined by experiments with the *rad54Δ cdc7-as3* strain (Figure 3C), in which cells are arrested after the meiotic S-phase, the latter is not the source of meiotic mutagenicity observed in *rad54∆* cells; rather, mutations appear in tight correlation with recombination events. We therefore conclude that the source of the elevated mutagenicity seen in *rad54Δ* mutants is triggered by DNA breakage and is associated with recombination.

Tid1 is a paralog of Rad54. Although they share structural homology, Tid1 and Rad54 have non-redundant functions [47,55]. Tid1 is needed to stimulate Dmc1 activity, and together, they act to preferentially support interhomolog recombination [23,56]. We found that in the *tid1∆* strain meiotic recombination dropped considerably to 35% (Figure 2B and [22]). Meiotic mutations decreased to 9% of the WT (Figure 2C and Table 2). From our findings, we suggest that the disruption of the Rad51–Rad54 pathway (in *rad54Δ* strain) leads to higher mutagenesis during meiosis. On the other hand, when the meiotic Dmc1 pathway becomes less functional (as in *tid1∆* cells), DSB repair is channeled to a less mutagenic track. Therefore, one may suggest that upon the deletion of *TID1*, DSB repair may rely mainly on the sister chromatid and that type of repair pathway is less mutagenic than interhomolog repair. Nevertheless, our results show opposite and complementary effects of these two paralogs (Rad54, Tid1), not only in meiotic recombinational repair but also in the levels of mutagenicity obtained during meiosis and in mitotically dividing cells.

Hed1 prevents functional cooperation between Rad54 and Rad51 during the HR process [26]. Therefore, in the absence of Hed1, the Rad51/Rad54 complex is more active in DSB repair. If active Rad54 is involved in generating less mutations during DSB repair (Figure 2C), *hed1Δ* cells should utilize more frequently the Rad51/Rad54 repair pathway and mutations should appear at a lower level than in WT cells. Indeed, the mutation rate in *hed1Δ* cells was 33% of the WT level (Figure 1B,C and Table 2), while recombination surrounding the *CAN1* reporter was similar to the WT level (Table 3). Deletion of *HED1* efficiently overcomes the DSB repair defects caused by the *dmc1*Δ deletion [25], akin to *RAD51* overexpression [57]. Along the same line, in our time course experiments of the double-mutant *hed1Δ dmc1Δ*, recombination rates were reduced to 12% of the WT level (Figure 1B and Table 2) and mutations at *CAN1* were lowered to 15% of the WT level (Figure 1C and Table 2). In addition, a moderate drop (of 38%) in recombination levels was observed in *hed1Δdmc1Δ Can^R^* colonies examined for the exchange of flanking markers (Table 3). Similarly, whole-genome sequencing of tetrads showed a mild reduction of about 25% in crossing overs in the *dmc1∆ hed1∆* mutant [58].

To draw a more global view, our findings indicate that recombinational repair pathways differ in the degree to which they generate mutations. We propose that Rad54 may have a moderating effect on mutagenicity in meiosis and in mitotically cycling cells. Since meiosis aims to create genetic diversity, and mutagenesis is another source of genetic variation, the inhibition of the Rad51/Rad54 pathway (by Hed1) during meiosis leads to the generation of more mutations, while in mitotically dividing cells, HR repair makes extensive use of the Rad51/Rad54 pathway and mutagenesis is minimized. The enhanced mutagenicity observed during meiosis reflects a fine balance between factors that enhance mutagenesis and factors that restrain it, such as Rad54.

### 4.2. Single-Stranded DNA May Enhance Meiotic Mutagenicity

Investigation of the *dmc1∆* single mutant is challenging, since *dmc1∆* cells arrest during meiosis and colonies are obtained only upon return to mitotic growth (RTG). Therefore, in *dmc1∆* cells, meiotic and mitotic repair are mixed in various proportions. Nevertheless, our exploration of mutation occurrence in *dmc1Δ* cells allows us to propose another source for meiotic mutagenicity; in our time course experiments, *dmc1∆* cells maintained full viability upon RTG, with reduced recombination, but showed a robust increase in mutagenicity (Figure 1B,C and Table 2). Yeast cells lacking *DMC1* fail to complete prophase I and arrest during meiosis with excessive amount of DSBs [39] but retain full viability upon RTG as diploids and do not generate spores [38]. What could trigger mutagenicity in *dmc1∆* cells?

Bishop et al. (1992) examined the appearance and repair of DSBs in *dmc1∆* cells and showed that DSBs are generated at the same time as in WT cells, but the broken DNA ends accumulate at a higher level and undergo additional 5′ end digestion, generating ssDNA beyond what is normally observed in the WT [39]. *dmc1* mutants accumulate DSBs with long ssDNA for a prolonged meiotic time, since they are not converted to mature JM structures [59]. Furthermore, Dmc1p normally forms a long nucleofilament that protects the ssDNA prior to invasion [60,61]. Excessive ssDNA tails (in the absence of *DMC1*) that are left exposed and do not invade the homologous partner during meiosis may be subjected to mutations, explaining the elevated mutagenicity in *dmc1Δ* cells. Early in vitro studies showed that ssDNA is more prone to the accumulation of mutations than dsDNA [62]. More recently, in vivo studies in yeast and in human cancer cell lines demonstrated hypermutability in ssDNA [63]. In yeast, the density of mutations in ssDNA can exceed 100–1000-fold over the rest of the double-stranded genome (reviewed in [64] and references therein). In addition, Tiemann-Boege and colleagues have shown that recombination hotspots are mutagenic in human sperm and speculated that the formation of ssDNA at methylated CpG sites is the main driver for de novo mutations [6].

Based on our findings in *dmc1∆* cells, we hypothesize that ssDNA associated with recombination could be another source for mutagenicity during meiosis. Mimitou et al. (2017) determined an average of 822 nts resection tail length around recombination hotspot centers in normal yeast cells undergoing meiosis. Given that ~ 150 DSBs are normally generated in a single cell going through meiosis, we have previously estimated that ~310,000 nucleotides of ssDNA are present in each yeast WT cell during meiosis, which is about 2.3% of the entire genome [53]. Excess ssDNA may result not only in increased mutagenesis but may also impact the spectrum of mutations obtained in various recombination sites. Particularly in *dmc1Δ* cells, the extra ssDNA may lead to increased loss-of-function *can1* mutants due to deletions and other rearrangements. It will be interesting to test the mutation signature and spectrum at the molecular level, comparing *dmc1Δ* to WT cells. Indeed, the low mutation rate seen in *hed1Δdmc1Δ* cells further supports this scenario; *HED1* deletion bypasses the meiotic arrest of *dmc1Δ* cells, in which ssDNA tails are presumably shorter and remain exposed for shorter periods, and therefore, the mutation rate in *hed1Δdmc1Δ* mutants is considerably lower.

Alternatively, it is possible that the usage of mixed HR repair pathways in *dmc1Δ* cells that are arrested for long periods during meiosis or some yet unknown mechanism, generates the elevated mutagenicity observed in cells lacking *DMC1*.

### 4.3. Is Meiotic Joint Molecule Resolution a Mutagenic Process?

In the current study, the only resolvase that was examined for involvement in meiotic mutagenesis was *MUS81*. The kinetics of mutagenicity in *mus81∆* cells were similar to those of the WT, with the mutation rate in *mus81∆* reaching maximal levels at 8 h (0.85 mutations per 1 million cells at the *CAN1* reporter), which is 71% relative to the WT level (Figure 2C and Table 2). However, despite six independent experiments performed with this mutant, we could not detect a statistically significant difference between *mus81∆* and WT cells. Mus81 is not the only resolvase operating in meiosis. In fact, in *S. cerevisiae*, meiotic crossover resolution is less dependent on Mus81/Mms4 [65]; *mus81∆* cells show only a modest CO reduction and Mus81 may act to process 3′ flaps of over-replication during meiotic recombination [66]. If the reduction in mutation rate (~30%) observed in *mus81∆* cells is a true result, then Mus81-mediated resolution is somewhat mutagenic. However, even if there is no difference between *mus81∆* and WT cells, it is still possible that the resolution that occurs via some other resolvase is mutagenic. It leaves open the issue whether the resolution of joint molecules is a mutagenic event during meiosis. Testing the Mlh1-Mlh3/Exo1 resolvase complex (which has a more central function in meiotic Holliday junction resolution) may be interesting but needs to be examined with caution, since *MLH1* or *MLH3* deletion is associated with several meiotic dysfunctions [67].

### 4.4. Concluding Remarks

Sexual reproduction and meiosis provide an effective mechanism to shuffle existing genetic information. We consider meiotic mutagenicity as another source of genetic diversity that passes on from parents to offspring. We provide, for the first time to our knowledge, experimental data showing that different recombination pathways disclose different mutagenicity loads during meiosis. We place these new findings regarding meiotic mutagenicity in the context of previous observations concerning recombination and DSBs repair. Mechanisms that enhance meiotic mutagenicity alter sequences in the genome of germ cells and may lead to alternations in allele frequencies in populations over evolutionary timescales.

## Figures and Tables

**Figure 1 genes-14-02017-f001:**
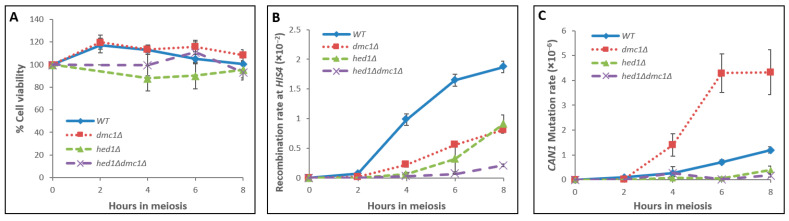
Kinetics of viability, recombination, and mutation events in WT, *dmc1Δ*, *hed1∆*, and double-mutant *hed1Δdmc1Δ* strains during meiosis. Aliquots of meiotic cells were taken at the times indicated and assayed for (**A**) cell viability (%) on YEPD plates, (**B**) recombination at *HIS4* on −His plates, and (**C**) mutation at *CAN1* on Can −Ade plates. WT (*n* = 19, in blue), *dmc1∆* (*n* = 8, in red), *hed1∆* (*n* = 9, in green), and *hed1∆dmc1∆* (*n* = 9, in purple). n denotes the number of independent experiments for each strain. Recombination rate—recombinations per 100 meiotic events. Mutation rate—mutations per million meiotic events. The number of colonies appearing on selection plates from time zero (either on −His or on Can −Ade plates) was always subtracted from the numbers obtained at later time points. Therefore, the net meiotic contribution at the beginning of the experiment is zero.

**Figure 2 genes-14-02017-f002:**
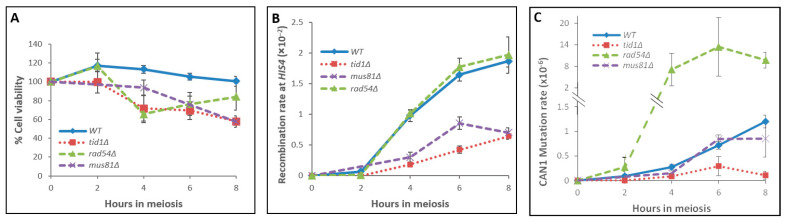
Kinetics of viability, recombination, and mutation events in WT, *rad54Δ*, *tid1∆*, and *mus81Δ* strains during meiosis. Aliquots of meiotic cells were taken at the times indicated, returned to vegetative growth, and assayed for (**A**) cell viability (%) on YEPD plates, (**B**) recombination at *HIS4* on −His plates, and (**C**) mutation at *CAN1* on Can −Ade plates. WT (*n* = 19, in blue), *tid1∆* (*n* = 7, in red), *rad54∆* (*n* = 7, in green), and *mus81∆* (*n* = 6, in purple). In C, the Y axis is split so that the lower half of the plot covers 0 to 1.5 and the upper half of the plot covers 2 to 20 regarding *CAN1* mutation rates ×10^−6^. n denotes the number of independent experiments for each strain. Recombination rate—recombinations per 100 meiotic events. Mutation rate—mutations per million meiotic events. The number of colonies appearing on selection plates from time zero (either on −His or on Can −Ade plates) was always subtracted from the numbers obtained at later time points. Therefore, the net meiotic contribution at the beginning of the experiment is zero.

**Figure 3 genes-14-02017-f003:**
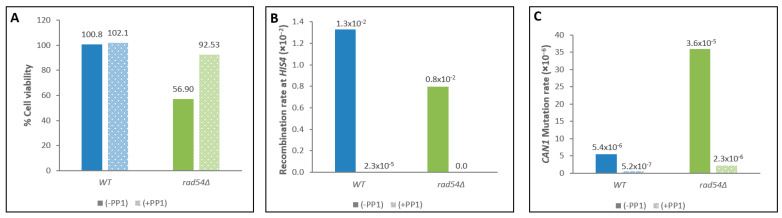
Viability, recombination, and mutations obtained at 8 h in WT and in *rad54*∆ cells upon treatment with the PP1 inhibitor. Aliquots of meiotic cells were taken at 8 h following transfer to SPM, treated with the PP1 inhibitor (dotted bars) or without PP1 (solid bars). (**A**) Cell viability (%) on YEPD plates. (**B**) Recombination at *HIS4* on −His plates. (**C**) Mutation at *CAN1* on Can −Ade plates. WT (blue); *rad54*∆ (green).

**Table 1 genes-14-02017-t001:** Yeast strains used and constructed in this study and their relevant genotypes.

Strain Name	Relevant Mutations	Genotype
YRA10	WT, *MAT***a**	*MAT***a**, *ho∆::LYS2*, *lys2*, *leu2*, *his4X*, *trp1∆::hisG*, *ura3*, *ade2∆::hisG*, *ADE2-CAN1* (*ADE2* inserted upstream of *CAN1*), *TRP1-LYP1* (*TRP1* inserted upstream of *LYP1*)
YRA12	WT, *MAT***α**	*MAT*α, *ho∆::LYS2*, *lys2*, *leu2*, *his4B::LEU2*, *trp1∆::hisG*, *ura3*, *ade2∆::hisG*, *avt2∆::URA3 (URA3* inserted downstream of *can1*), *can1::kanMX6*, *lyp1::natNT2*
YRA23-1	diploid WT, *CDC7*	YRA10 × YRA12
YRA17	*spo11∆*	YRA23-1 with homozygous *spo11∆::hisG*
YRA34-1	*mre11∆*	YRA23-1 with homozygous *mre11∆::kanMX4*
YRA20	*dmc1∆*	YRA23-1 with homozygous *dmc1∆::kanMX4*
YRA55	*hed1∆*	YRA23-1 with homozygous *hed1∆::kanMX4 (ydr015c)*
YRA56	*hed1∆ dmc1∆*	YRA23-1 with homozygous *hed1∆::kanMX4 (ydr015c)*, *dmc1∆::kanMX4*
#3512	*rad54∆*, *MAT***a**	*MAT***a**, *ho∆::LYS2*, *lys2*, *ura3*, *leu2∆::hisG*, *his4X::LEU2-BamHI-URA3*, *rad54∆::URA3*
ALY77	*rad54∆*, *MAT***a**	*MAT***a**, *ho∆::LYS2*, *lys2*, *leu2*, *his4X*, *trp1∆::hisG*, *ura3*, *ade2:∆::kanMX4*, *ADE2-CAN1* (*ADE2* inserted upstream of *CAN1*), *TRP1-LYP1* (*TRP1* inserted upstream of *LYP1*), *rad54∆::URA3*
ALY92	*rad54∆*, *MAT***α**	*MAT***α**, *ho∆::LYS2*, *lys2*, *leu2*, *his4B::LEU2*, *trp1∆::hisG*, *ura3*, *ade2∆::kanMX4*, *avt2∆::URA3* (*URA3* inserted downstream of *can1*), *can1::kanMX6*, *lyp1::natNT2*, *rad54∆::URA3*
ALY94	diploid, *rad54∆*	ALY77 × ALY92
YAS3	*tid1∆*	YRA23-1 with homozygous *tid1∆::kanMX4*
YRA51	*mus81∆*	YRA23-1 with homozygous *mus81∆::kanMX4*
OMY78-1	WT, *MAT***a**, *cdc7-as3-9myc*	*MAT***a**, *hoΔ::LYS2*, *leu2Δ::hisG*, *his4x*, *lys2*, *trp1∆::hygB*, *ura3∆::hisG*, *ade2∆::kanMX4*, *cdc7-as3-9myc*, *ADE2-CAN1* (*ADE2* inserted upstream of *CAN1*), *TRP1-LYP1* (*TRP1* inserted upstream of *LYP1*)
OMY75-2	WT, *MAT***α**, *cdc7-as3-9myc*	*MAT***α**, *ho∆::LYS2*, *ade2∆::kanMX4*, *trp1∆::hph*, *his4B::LEU2*, *avt2∆::URA3* (*URA3* inserted downstream of *can1*), *cdc7-as3-9myc*, *can1::kanMX6*, *lyp1::natNT2*
OMY93	diploid WT, *cdc7-ac3-9myc*	OMY78-1 × OMY75-2
ALY170	*rad54∆*, *cdc7-as3-9myc*	ALY94 with homozygous *cdc7-as3-9myc*

In *can1*^R^::*kanMX6* allele, the promoter and the first six codons of the ORF of the *CAN1* gene are replaced by *kanMX6.* In the *lyp1^R^::natNT2* allele, the promoter and the first six codons of the ORF of the *LYP1* gene are replaced by *kanMX6.*

**Table 2 genes-14-02017-t002:** Summary of viability, spore germination, recombination, and mutations obtained at 8 h in WT and in various mutants examined in the current study.

Mutated Gene	% Cell Viability (at 8 h)	Spore Germination	Recombination at *HIS4* (at 8 h) (×10^−2^)	Mutation at *CAN1* (8 h)	
				Meiotic Mutations (×10^−6^)	Fold Change Mutant/ WT
WT	100	96% (366/380)	1.87	1.20	1.00
*spo11∆*	36	0% (0/150)	0.00	0.12	0.10
*mre11∆*	56	0% (0/146)	0.01	0.24	0.20
*dmc1∆*	108	Φ	0.81	4.33	3.60
*hed1∆*	96	93% (164/176)	0.91	0.39	0.33
*hed1∆dmc1∆*	93	55% (173/312)	0.22	0.18	0.15
*rad54∆*	84	66% (105/160)	1.97 *	9.70	8.07
*tid1∆*	58	N.D	0.65	0.11	0.09
*mus81∆*	58	56% (158/280)	0.71	0.85 *	0.71

From left to right: cell viability, spore germination, recombination frequencies at *HIS4*, and mutation rates at *CAN1*. * Results shown with asterisk (*) do not differ from the WT and are not statistically significant. All other values presented in this table are statistically significant (with a *p* value less than 0.05). Φ spores are not generated due to meiotic arrest of *dmc1Δ* cells.

**Table 3 genes-14-02017-t003:** % Recombinants of markers flanking the *CAN1* reporter in *Can^R^*-mutated colonies of WT, *hed1∆*, *hed1∆dmc1∆*, and *mus81∆* strains.

Mutated Gene	Strain	Recombinants around *CAN1* Reporter in *CAN*^R^ Mutants	Fold Change Mutant/WT
WT	YRA23-1	39% (95/243)	1
*hed1∆*	YRA55	40% (69/171)	1
*hed1∆dmc1∆*	YRA56	25% (38/153)	0.64
*mus81∆*	YRA51	15% (34/229)	0.38

Sporulating cultures of WT, *hed1∆*, *hed1∆dmc1∆*, and *mus81∆* from SPO plates were placed on YPD plates, and four-spore asci were dissected by micromanipulation. YPD plates were incubated at 30 °C for 4 days, and spore germination was assessed. Germinated spores were then replica-plated onto -Ura and –Ade plates to examine recombination around the *CAN1* reporter. The numbers in parentheses show recombinant colonies out of the number of *Can^R^* haploid colonies. Fold change (compared to the WT) of recombination around the *CAN1* reporter is shown on the right.

**Table 4 genes-14-02017-t004:** Spontaneous mitotic mutation rate at the *CAN1* reporter gene in WT, *rad54∆*, and in *tid1∆* strains.

Mutated Gene	Strain	Mitotic Mutation Rate at *CAN1* (×10^−6^) In Mitotic Cells	Fold Change Mutant/WT
WT	YRA10	0.2	1
*rad54∆*	ALY94	5.14	25.7
*tid1∆*	YAS3	0.3	1.5

Each value (×10^−6^) represents the average of at least four independent experiments. All values are statistically significant according to a *t*-test with a *p* value less than 0.05. Fold change (compared to the WT) of mutation rate in vegetatively cycling cells is denoted on the right.

## Data Availability

Not applicable.

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
