# Peer review of "Homologous Recombination and Repair Functions Required for Mutagenicity during Yeast Meiosis"

_genes, 2023, doi:10.3390/genes14112017_

Round 1

Reviewer 1 Report

Morciano and coworkers extend what is known about recombination-induced mutagenesis in meiosis, by examining the effects of various mutants that affect the meiotic recombination process, using a selection for loss-of-function in the CAN1 gene when meiotic cells are returned to vegetative growth. Thus, recombination and mutation rates measured reflect the sum of events actually occurring during meiosis and those that were initiated during meiosis and are completed during return to growth. The authors convincingly show that meiotic mutagenesis does not occur during meiotic S-phase, and that it requires the Spo11-dependent DNA double-strand breaks that initiate meiotic recombination.

The main findings of the paper are that both dmc1∆ and rad54∆ mutants display mutation rates substantially greater than wild type (about 4- and 10-fold, respectively). Interestingly, in hed1∆ mutants, where Rad54-dependent recombination is increased, mutagenesis is decreased. Furthermore, loss of Tid1, a translocatase related to Rad54, results in reduced mutagenesis. This is all consistent with the suggestion that Rad54-dependent repair processes help prevent mutagenesis, a finding with implications for mechanisms of meiotic double-strand break-repair in the presence or absence of Rad54.

The manuscript is written very well, and the presentation and analysis of the data and procedures are clear and straightforward. The paper will be of interest to a broad spectrum of researchers studying DNA repair, homologous recombination, and meiosis. It is suitable for publication as is, but I have two minor suggestions:

1. Figure 2 panel C uses one Y axis (linear) for wild type, tid1∆, and mus81∆, and a second Y axis (log scale) for rad54∆. This makes comparison of data difficult, especially at the 2h timepoints referred to below (lines 384-5). I suggest using a single split-Y axis, so that the lower half of the plot covers 0 to 1.5, and the upper half of the plot covers a larger range (say, 2.5-25). 

2. In the consideration of the relationship between ssDNA amounts and mutagenicity (lines 528-566), it should be considered that excess ssDNA may result not only in increased mutagenesis but a change in mutation spectrum; all that extra ssDNA may lead, for example, to increased loss-of-function can1 mutants due to deletions and other rearrangements. Since an examination of can1 mutant spectra is beyond the scope of this paper, the discussion should be modified to include this possibility.

Author Response

We thank reviewer No 1. for thoroughly reading the manuscript and for your  insightful suggestions. See, below, our answers to your comments, one by one (in purple). 

  1. Figure 2 panel C uses one Y axis (linear) for wild type, tid1∆, and mus81∆, and a second Y axis (log scale) for rad54∆. This makes comparison of data difficult, especially at the 2h timepoints referred to below (lines 384-5). I suggest using a single split-Y axis, so that the lower half of the plot covers 0 to 1.5, and the upper half of the plot covers a larger range (say, 2.5-25). 

This is a very good advice, to split the Y axis into 2 separate scales, since it allows the reader to examine in more detail the mutants’ behavior, especially at the 2 hours timepoint. Indeed, we now modified figure 2 C accordingly, so that the lower half of the plot covers 0-1.5 and the upper half of the plot covers 2-22. We added an explanation to the figure legend, lines 387-388 (in the corrected version), as follows: “The Y axis is split, so that the lower half of the plot covers 0 to 1.5 and the upper half of the plot covers 2 to 20, regarding CAN1 mutation rate x10-6

  1. In the consideration of the relationship between ssDNA amounts and mutagenicity (lines 528-566), it should be considered that excess ssDNA may result not only in increased mutagenesis but a change in mutation spectrum; all that extra ssDNA may lead, for example, to increased loss-of-function can1mutants due to deletions and other rearrangements. Since an examination of can1mutant spectra is beyond the scope of this paper, the discussion should be modified to include this possibility.

We thank reviewer No 1. for this good suggestion. We modified the text, accordingly, emphasizing that mutation signature (not only increase mutagenesis) may be affected in dmc1D cells. Please see the addition we made in lines 575-579 (in the corrected version), as follows: “Excess ssDNA may result not only in increased mutagenesis but may also impact the spectrum of mutations obtained in various recombination sites. Particularly in dmc1D  cells, the extra ssDNA may lead to increased loss-of-function can1 mutants due to deletions and other rearrangements. It will be interesting to test mutation signature and spectrum on the molecular level, comparing dmc1D to WT cells “

Reviewer 2 Report

Morciano et al., explores mutagenesis during meiosis and HR genes’s involvement in it. The authors show that the HR pathway and DNA repair processes that are associated with meiotic recombination are mutagenic providing an additional source of variability that can contribute to evolution.

Comments:

Do you have evidence of correlation between meiotic hotspots to  increased sequence variability?

It would have been good to show DSBs or ssDNA directly in Dmc1 , hed1,  Dmc1 hed1 mutants.

Different fonts have been used throughout the manuscript as if copy pasted. Please change these to one font.

Lack of punctuations at multiple places. For eg Line 259.

Sentences like “We wished to confirm” at line 386 doesn’t make sense. You are testing a hypothesis. Please write the manuscript appropriately.

There are grammatical mistakes that needs to be corrected.

Decision:

Accept after these comments/concerns being addressed.

Must be improved

Author Response

We thank reviewer No 2. for thoroughly reading the manuscript and for your  good suggestions. See, below, our answers to your comments, one by one (in purple). 

Do you have evidence of correlation between meiotic hotspots to increased sequence variability?

No, we don’t have evidence for such correlation. We very much agree that examining changes in sequences around meiotic hotspots is an important step towards understanding mutation signature in meiotic cells, namely mutations that pass from one generation to the next. Indeed, this article sets the stage for such an examination; In the current study, we analyzed several relevant Homologous repair gene-deletions and found that both rad54D and dmc1D cells show elevated mutagenicity during meiosis. Based on our findings in the current study, our ongoing research and maybe other laboratories carry this topic further and analyze mutation spectrum and mutation signature in the above-mentioned mutants and compare it to WT. To emphasize this to the reader, we inserted in the text the following sentence: “It will be interesting to test mutation signature and spectrum on the molecular level, comparing dmc1Δ to WT cells.” (lines 569-570 in the corrected version). Tracking sequence variability is out of the scope of this article, but definitely worth examining in the future.

In addition, Tiemann-Boege and colleagues (Arbeithuber et al. PNAS 2015) directly sequenced many single sperm DNA molecules and found that new mutations reside more frequently in molecules with a crossover than in molecules without a recombination event. This interesting finding encourages more research regarding sequence variability in relation to meiotic recombination hotspots in human, in S. cerevisiae and in other organisms.

It would have been good to show DSBs or ssDNA directly in Dmc1, hed1, Dmc1 hed1 mutants.

As the generation of DSBs and its repair were already shown for dmc1D (Bishop et al 1992. Cell, Zenvirth et al 1997. Genes to Cells), for hed1D and for dmc1D hed1D mutants (Tsubouchi H & Roeder GS. 2006. Genes & Development) in strains from the SK1 genetic background (the same as our strains), we decided to rely on these previous data and not test it again in the current study. Thus we connect meiotic DSBs repair to our observations in the dmc1D hed1D mutant (lines 509-514 in the corrected version). To emphasize the connection between DSB repair and our observations regarding meiotic mutagenicity in dmc1-deleted cells, we added the following sentence “Yeast cells lacking DMC1 fail to complete prophase I, halt during meiosis with excessive amount of DSBs… “(lines 540-541 in the corrected version).

As for ssDNA, in dmc1D strains it was already shown that cells are arresting in meiosis with longer starches of single-stranded DNA (Bishop et al 1992. Cell). We mentioned this important phenomenon in the text as follows “…dmc1∆ cells and showed that DSBs are generated at the same time as in WT cells, but the broken DNA ends accumulate at a higher level and undergo additional 5' end digestion, generating ssDNA beyond what is normally observed in WT” (lines 544-547 in the corrected version).

Different fonts have been used throughout the manuscript as if copy pasted. Please change these to one font.

The whole manuscript was read and revised by the authors and all fonts were unified to the “Palatino Linotype“ style, as required by MDPI Genes.

Lack of punctuations at multiple places. For eg Line 259.

We carefully scanned through the manuscript and tracked various places where punctuations and comas were missing, as follows:

Additional punctuations were introduced in appropriate places such as:  lines: 51, 265, 416, 433, 557

We added commas in many places, such as in lines: 15, 22, 48, 65,215, 257, 291, 319, 355, 409 471, 546, 570

Sentences like “We wished to confirm” at line 386 doesn’t make sense. You are testing a hypothesis. Please write the manuscript appropriately.

We agree with the reviewer and modified the sentence as follows: “We tested the hypothesis that meiotic S-phase is not the source for the elevated meiotic mutation levels observed in the rad54Δ mutant“(lines 391-392 in the corrected version).

There are grammatical mistakes that needs to be corrected. 

Thanks to the reviewer’s attention, we revised the manuscript and introduced many grammatical (and other) changes, such as in lines: 24-26, 37-38, 57-58, 401, 437, 458, 546, 563,581-582, and in other places in the corrected version of the MS.

While revising the manuscript, we paid special attention to reviewer No.2 comments and tried to improve the description of the Methods (as in lines 123, 212-213, 219-220 and in other places in the corrected version of the Materials and Methods section).

We also introduced changes in the Results section to address the reviewer’s concern and clarified the presentation of the results. For example, see lines: 250-253, 270-272, 388-390, 421-423, and in some other places in the corrected version of the Results section.